# Domestic Cat Sound Classification Using Learned Features from Deep Neural Nets

## Yagya Raj Pandeya, Dongwhoon Kim and Joonwhoan Lee *

Division of Computer Science and Engineering, Chonbuk National University, Jeonju 54896, Korea;
yagyapandeya@gmail.com (Y.R.P.); clickmiss123@naver.com (D.K.)
* Correspondence: chlee@chonbuk.ac.kr

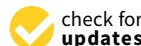

**Featured Application: Domestic cats are ancient human pet animal that communicate through generating sounds. Automatic animal sound classification creates a better human-animal communication environment that will obviously be helpful to know the pet animal requirements more clearly. Sometime pet also protect human being in case of natural disaster or in criminal activity by alerting messages. In this work, our main contribution is domestic cat sound classification using limited number of available data. We try to overcome the lack of data problem using different well-known techniques such as transfer learning, ensemble, cross validation, etc. We propose frequency division average pooling (FDAP) technique instead of global average pooling (GAP) to make a robust prediction using various frequency band features. In order to better understand our results, we visualize networks learned features, confusion matrix, and receiver operating characteristic (ROC) curve. The proposed techniques to deal with lack of data problem for domestic cat sound classification can be useful to other researchers working in similar domain.**

**Abstract:** The domestic cat (*Feliscatus*) is one of the most attractive pets in the world, and it generates mysterious kinds of sound according to its mood and situation. In this paper, we deal with the automatic classification of cat sounds using machine learning. Machine learning approach for the classification requires class labeled data, so our work starts with building a small dataset named *CatSound* across 10 categories. Along with the original dataset, we increase the amount of data with various audio data augmentation methods to help our classification task. In this study, we use two types of learned features from deep neural networks; one from a pre-trained convolutional neural net (CNN) on music data by transfer learning and the other from unsupervised convolutional deep belief network that is (CDBN) solely trained on a collected set of cat sounds. In addition to conventional GAP, we propose an effective pooling method called FDAP to explore a number of meaningful features. In FDAP, the frequency dimension is roughly divided and then the average pooling is applied in each division. For the classification, we exploited five different machine learning algorithms and an ensemble of them. We compare the classification performances with respect following factors: the amount of data increased by augmentation, the learned features from pre-trained CNN or unsupervised CDBN, conventional GAP or FDAP, and the machine learning algorithms used for the classification. As expected, the proposed FDAP features with larger amount of data increased by augmentation combined with the ensemble approach have produced the best accuracy. Moreover, both learned features from pre-trained CNN and unsupervised CDBN produce good results in the experiment. Therefore, with the combination of all those positive factors, we obtained the best result of 91.13% in accuracy, 0.91 in f1-score, and 0.995 in area under the curve (AUC) score.

**Keywords:** balanced dataset; data augmentation; deep belief network; feature extraction; frequency division average pooling; ensemble

---

## 1. Introduction

The sound generation and perception systems of animals have evolved to help them to survive in their environment. From an evolutionary perspective, the intentional sounds that are generated by animals should be distinct from the random sounds of the environment. Some animals have special sensory capabilities, such as vision, sights, feeling, and awareness of natural changes as compared to human beings. Animal sounds can be helpful for human beings in terms of security, prediction of natural disasters, and intimate interactions if we are able to recognize them properly.

The data-driven machine learning approach for acoustic signal has been of great interest to researchers in recent years, and some studies have been conducted on animal sound classification [1,2] and animal population identification based on their sound characteristics [3]. The marine mammal sound classification and its impact on marine life have been studied in [4,5]. Several studies [6–12] have focused on bird sound identification, classification, and its challenges. The study in [13] has performed the insect species classification based on their sound signals. The prediction of unusual animal sound behavior during the earthquake and natural disaster has been studied using machine learning techniques in [14]. Recently, the possibility of transfer learning from music features for bird sound identification using hidden Markov models (HMMs) has been examined in [15].

The pet animals are close friends of the human from the time of human evolution, and they deliver their messages by producing some identical sounds. Most pets spend their whole time in human peripheries, thus sound analysis of pet animals is important. Domestic cat is one of the most widely loved pet animals in the world, and the whole population is around 88.3 million, according to the report of Live Science (2013). The behavioral analysis of domestic cat is explained well in [16,17].

The attempt for environmental sound recognition using HMMs [18] shows classification of 15-classes of various environmental sounds. Among 15-classes also include animal sounds (bird call, dog barking, and cat meowing) and shows better sound recognition accuracy using universal modeling based on GMM clustering rather than simple HMM and balanced universal modeling. The domestic cat sound classification using transfer learning [19] is a primary step to classify cat sounds with limited data. The classification of cat sounds is an attempt to communicate better with domestic cat, and hence possibly to understand their intentions well.

This paper is an extension of [19] that particularly focuses on dealing with small datasets for automatic classification of domestic cat sounds using machine learning algorithms. The data driven approaches requires class labeled data, therefore our work starts with building a small set of cat sound dataset. Our goal is to achieve a general purpose classification system for domestic cat that is independent from the issues, such as the bio-diversity, species variation, and age variation, so we try to include many kinds of cat sounds in different moods. Because the set of collected data is not enough, we increase the amount of data using various audio augmentation methods. As in [20], audio augmentation is performed by random selection of augmentation methods within the appropriate ranges of parameter values, as described more in Section 3.1.

As there is little knowledge about which audio features are efficient for our purpose, instead of hand-crafted features, we use two kinds of learned features from deep neural networks: convolutional neural network (CNN) and convolutional deep belief network (CDBN). The CNN features are obtained from pre-trained network on music data by transfer learning. Meanwhile, the unsupervised CDBN network features are extracted from the network solely trained on a collected set of cat sounds without class labels. We compare the two networks with their learned features in terms of classification performance of the cat sounds.

We try to classify the cat sounds using five different machine learning algorithms, and finally ensemble their predictions to make the performance more robust. Each of these five algorithms is powerful for classification problems and needs less labeled data for training because of the small number of parameters. The cat sounds in each class have different nature, which mainly varies in frequency bands. To make our classification more efficient, we also modify the conventional global average pooling for dimension reduction, and name it frequency division average pooling (FDAP).

The FDAP first divides each feature map in different frequency bands and take the average value in each band.

For the classification we exploit five different machine learning algorithms and an ensemble of them. We compare the classification performances with respect to the following factors; the amount of data increased by augmentation, the learned features from pre-trained CNN or unsupervised CDBN, conventional global average pooling (GAP) or FDAP, the five different machine learning algorithms, and an ensemble of them. As we expected, the FDAP features with the data increased using augmentation produces the best accuracy in the ensemble approach. Also both learned features from pre-trained CNN and unsupervised CDBN produce successful results in the experiment. Therefore, we have obtained with the combination of all those positive factors the best result of 91.13% in accuracy, 0.91 in f1-score, and 0.995 in area under the curve (AUC) score.

This paper is organized in the following order. Section 2 describes the domestic cat sound dataset preparation. Section 3 is an overview of data pre-processing, CNN and CDBN network feature extraction, and classification of the extracted features. The results with discussion are included in Section 4. Finally, the conclusion and future works are mentioned in Section 5.

## 2. Cat Sound Dataset

Hearing is the second most important human sense after vision to recognize any animal. Automatic cat sound recognition using data driven machine learning needs a large amount of labeled data for successful training. It is a great challenge to collect domestic cat sounds. The Kaggle (https://www.kaggle.com) challenge *Audio Cats and Dogs* provides 274 cat sound audio files but they do not have any categories of the sounds. Because there is no large sound dataset for the domestic cat, we have collected them mostly from online videos source of YouTube (https://www.youtube.com/) and Flicker (https://www.flickr.com/).

The pie chart in Figure 1 illustrates cat sound dataset, named as *CatSound dataset*, where each class is named according to the sound of cats in various situations. The number of cat sound files in each class is nearly 300, which is about ten percent of the total number of data files, so that our *CatSound dataset* is balanced. The total duration of *CatSound dataset* is more than 3 h for all sounds in 10 classes, which implies that the average duration of a sound is about 4 s.

The proper categorization of the collected sounds was another big challenge, because some sounds in one category are very similar to others in different categories. In addition, cats often produce different sounds with small-time difference. For example, an angry cat might generate "*growling*", "*hissing*", or "*nyaaan*" sounds almost simultaneously. Semantic explanation of the domestic cat sounds in [21,22] helped us to categorize the various sounds into proper classes.

Some audio files in our dataset have been divided into several segments with varying length, each of which has the same semantic category in the same way used in [23] for music information retrieval. For example, the sound of the cat in normal mood ("*meow-meow*"), defensing ("*hissing*"), a kitten calling its mother ("*pilling*"), and cats in pain ("*miyoou*") can be semantically correct, even in short time duration. On the other hand, the cat sounds in rest ("*purring*"), warning ("*growling*"), mating ("*gay-gay-gay*"), fighting ("*nyaaan*"), angry ("*momo-mooh*"), and want-to-hunt ("*trilling* or *chatting*") are usually more meaningful if they are analyzed in longer time duration. Even in some classes of cat sound, the data may have varying length because the bio-diversity widely differs depending on geographical locations, cat species, and ages. The waveform representations of 10 samples from each class are visualized in Figure 2. The pictorial representation helps the reader to understand nature, amplitudes and time duration of the sounds.

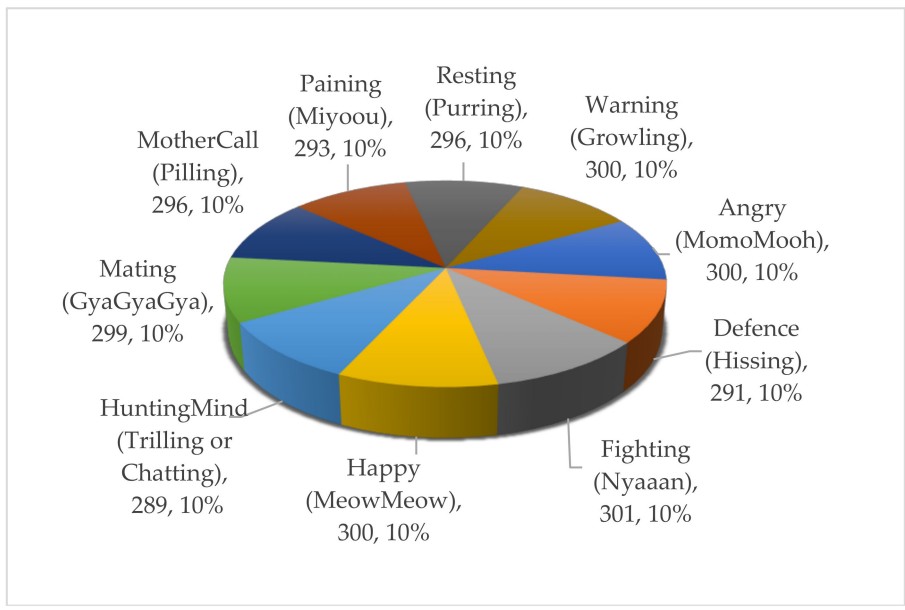

**Figure 1.** Balanced cat sound dataset representation. Each class in the dataset is named according to the mood or the situation of the cat when they are making sound. The onomatopoeia words in parentheses are the imitation of the cat sounds in each class. The numbers indicate the number of cat sounds and the numbers with percent is the corresponding portion of each class size in the dataset.

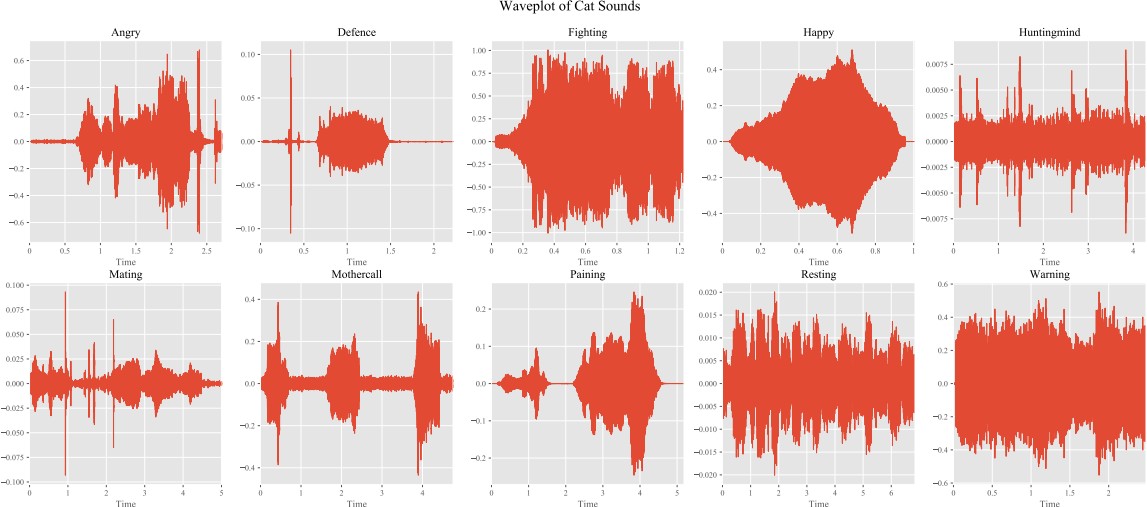

**Figure 2.** Waveform visualization of 10 data sample taken from each class of *CatSound dataset.*

## 3. Method for Classification

This section covers the data augmentation and pre-processing, transfer learning, frequency division average pooling, and feature extraction of our dataset. This section also discusses various classifiers and the ensemble methods that are used in our work.

### 3.1. Data Augmentation and Pre-Processing

Since the *CatSound dataset* is too small to train the machine learning algorithms, it is necessary to increase the size of the data using proper data augmentation. In the experiment, we have performed the audio augmentation, as described in [20], using random selection of augmentation methods with their parameters within the appropriate ranges. The augmentation methods include speed change (range 0.9 to 1), pitch shifting (range −4 to 4), dynamic range compression (range 0.5 to 1.1), insertion of noise (range 0.1 to 0.5), and time shifting (20% in forward or backward direction). Note that each

parameter has to be chosen in the modest range, otherwise the sound can be drastically changed and categorized differently from the original sound.

Three kinds of augmented datasets, namely 1x_Aug, 2x_Aug, and 3x_Aug, and one original dataset of cat sounds are used in the experiment. The 1x_Aug dataset contains the original data and its augmented clone. Similarly, the 2x_Aug, and 3x_Aug dataset consist of the original audio file, and its additional augmented clones with doubled and tripled in size, respectively. These four datasets (one original and three augmented) are prepared for the input of the pre-trained CNN and unsupervised CDBN for feature extraction.

The data preprocessing for each of the two neural networks for feature extraction is different because of the structure difference. The first step of our input signal processing is zero padding to make the full length audio that can generate the fixed size mel spectrogram. The role of zero padding in the time domain signal is to increase the frequency resolution and make full periods in signal that remove the spectral leakage. The sampled cat sound audio signals with 16 kHz rate is represented by mel spectrogram as input for both networks. For the CNN, the mel spectrograms are extracted from each cat sound dataset using Kapre [24]. The input to the CNN has a single channel, 96-mel bins, and 1813 temporal frames. In the case of CDBN, the mel spectrogram with 96-mel bins and 155 temporal frames is prepared for the input after whitening. The preprocessing procedures for both the CNN and CDBN network are given in Figure 3.

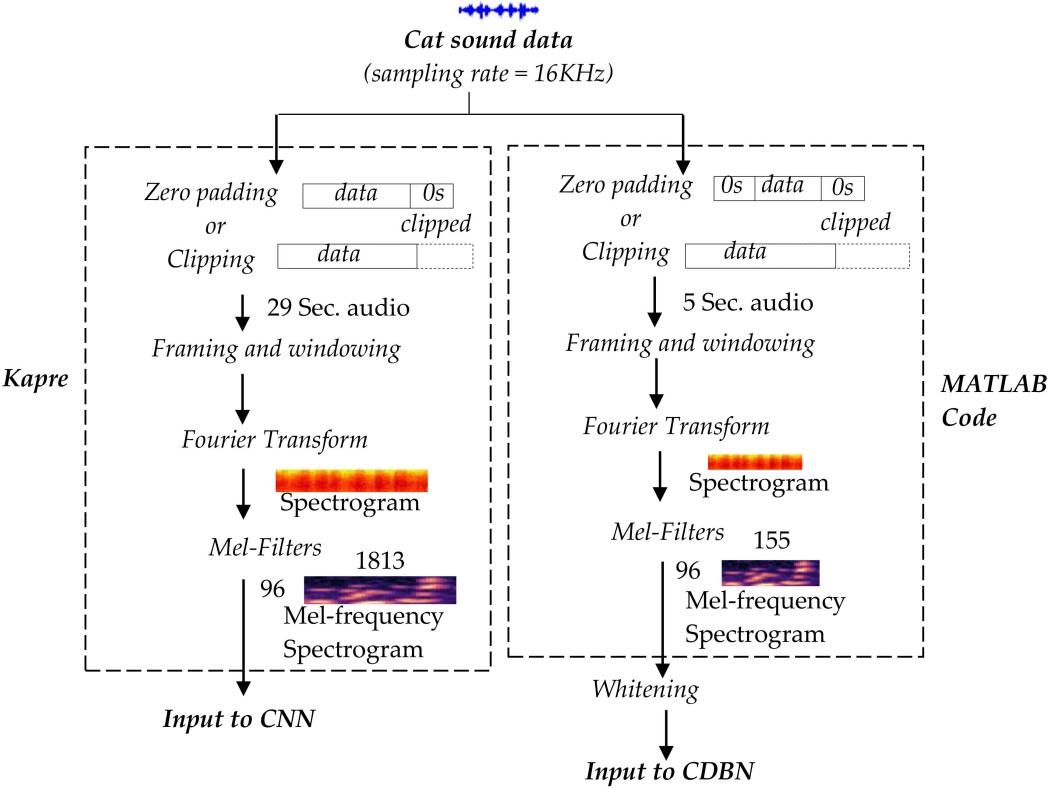

**Figure 3.** Preprocessing of the input data for the convolutional neural network (CNN) (**left**) and convolutional deep belief network (CDBN) (**right**). For the CNN input data, all the preprocessing is made using Kepre and short audio data are padded with zeros only on the right side. The MATLAB code is used for CDBN input preprocessing and the short audio data are padded with zeros on both sides. Finally, whitening is carried out at the end.

### 3.2. Frequency Division Average Pooling

The conventional GAP layers vectorize the feature maps of the convolutional network, as described in [25]. GAP is a dimension reduction technique that reduces a three-dimensional tensor into

one feature vector by taking the average of each feature map. For example, if a tensor has dimension of $H \times W \times C$, then GAP reduces the size to $1 \times 1 \times C$, where $H$, $W$, $C$ are the height, the width, and the channel of a tensor, respectively. One major advantage of using GAP is the reduction of network parameters because there are no parameters to optimize.

In this work, we used frequency division average pooling, which is modified version of GAP. This is an area specific feature extraction, where we first divide the feature map into low and high frequency bands and then apply the GAP in each band. This is an effective way to deal with frequency varying data, such as cat sounds, in which the frequency components in a certain band are active for some specific sound classes. The log power spectrogram and mel spectrogram of the sample data taken from each class of *CatSound dataset*, as illustrated in Figures 4 and 5, indicate that the activities of the cat sound in frequency bands are different depending on the sound class. Therefore, if the features are properly segmented in multiple frequency bands, we can expect the better classification performance. However, it is hard to know the exact cut-off points to divide feature maps into frequency bands, so we overlap the frequency division, like overlapping windows of short time Fourier Transform (STFT). The features are taken by overlapping small portion (single mel bin) of adjacent frequency components. There may be many alternative ways to divide features maps of networks into frequency bands. We divide the feature maps according to the number of available frequency components in each layer, and therefore, a higher layer feature map has relatively less number of divisions.

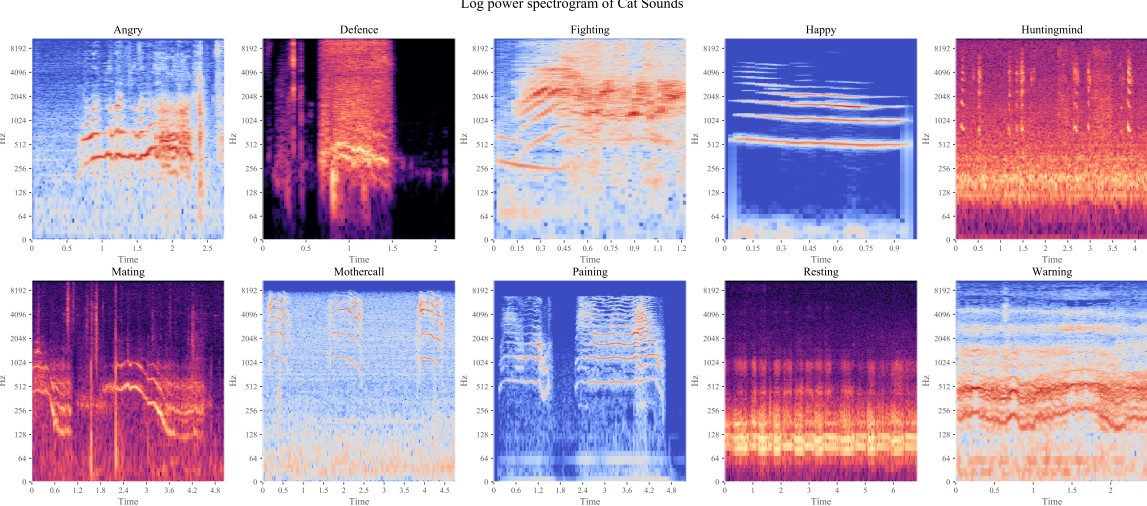

**Figure 4.** The sample in Figure 2 transformed to log power spectrogram.

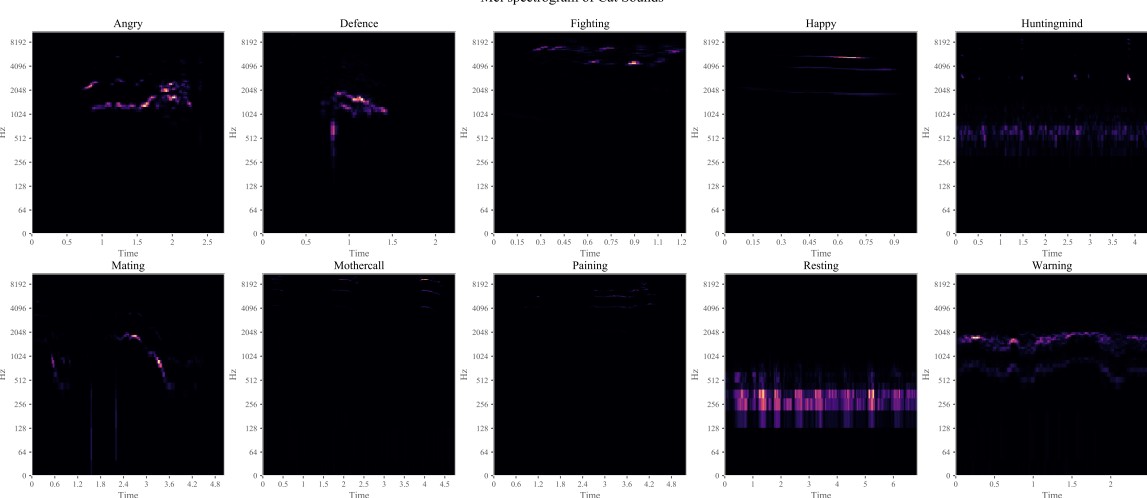

**Figure 5.** The sample in Figure 2 transformed to mel spectrogram.

### 3.3. Transfer Learning of CNN

Transfer learning is a transformation process of any learned knowledge from one or more source domains to a target domain, so that the target may generalize its prediction capability to solve the similar problems. A key advantage of transfer learning is a boost of the performance when there is only a small number of labeled training dataset. This is possible if the source and target network have similar tasks. A recent research on transfer learning in music onset detection [26] shows that the difference in source and target datasets may cause a failure to capture the relevant information while using transfer learning. The transfer learning technique has been successfully employed and investigated in various research areas such as image classification [27,28], acoustic event detection [29,30], speech and language processing [31], and music classification [32,33].

In this work, however, there is no useful alternative to compensate the lack of data, except the transfer learning. Therefore, the pre-trained CNN with Million song dataset [34] is adopted as the feature extractor, and we check the efficiency of the learned feature from music for cat sound classification. The features from the network [33] by transfer learning shows good performance in music classification and regression. Although the cat sounds have some variation from the studio recorded music in terms of the frequency variation, signal-to-noise ratio, abrupt change in magnitude, and frequent interruption of environmental sounds, but some following studies show the close relation of music to animal sound. In western classic music, composers and musicians commonly use bird song in their music [15,35,36] in different ways [37]. Beside that the pet animals and wild animal sound are also included in songs [36] directly and indirectly for making background noises in concert halls [35]. A review study [38] illustrates close relationships among human languages, (human) music, and animal vocalization. As a conclusion, the diversified music in million song dataset has some relationship with cat sound and it can be one solution to deal with the lack of data issue in cat sound classification if transfer learning is performed. Hence, we use the pre-trained CNN as the source network for transfer learning.

We extracted features from this pre-trained network for both the original and augmented dataset using Keras (https://keras.io/). The high dimensions of extracted features in each layer are reduced down to a vector using FDAP method and then concatenated. The whole experiments have been performed with NVIDIA GeForce GTX 1080 Ti GPU. We encountered a memory problem when extracting the features from the first two layers, and hence, we inevitably used the conventional global average pooling in these two layers. To apply the FDAP in the third and fourth layer, the feature maps are divided equally into four bands. But, the feature maps of the fifth layer are divided into two overlapping bands, because there are only a small number of bins in the mel frequency dimension. There are 32 features in each band and all are concatenated to make the input vector for the five classifiers. The feature extraction and classification using the pre-trained CNN architecture is shown in Figure 6. The experimental results show that, even though the cat sound and music are very different, the audio signals share similar features, so that the transfer learning is helpful.

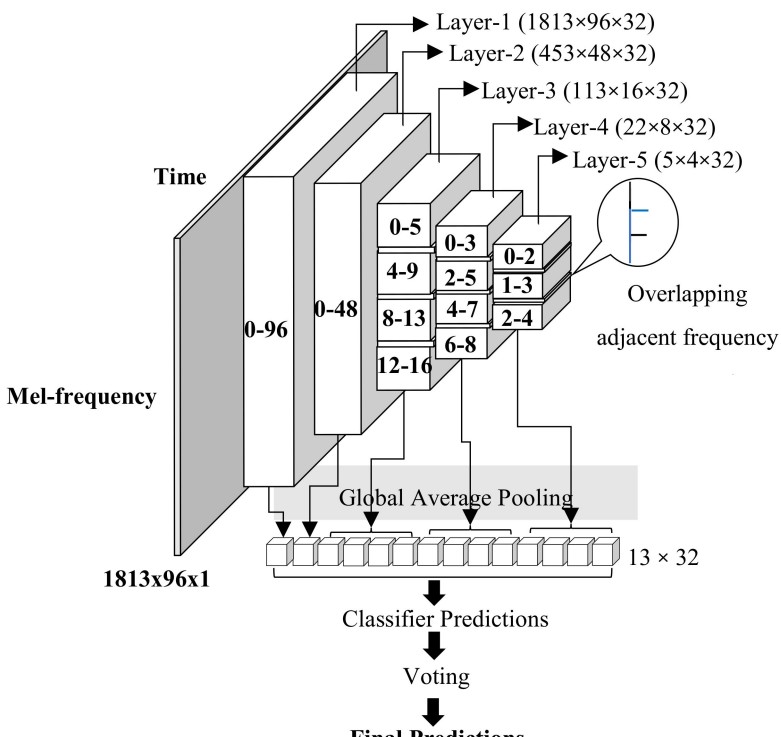

**Figure 6.** Transfer learning of CNN and the classification by extracted features. From layer one to five, the number of segments for frequency division average pooling (FDAP) in each layer is 1, 1, 4, 4, and 3, respectively. Each layer parameters are represented as (*time × frequency × channel*). Each segment produces a 1 × 32-dimensional feature vector, and the concatenated feature vector is fed into various classifiers and voting for the predicted probability of the final ensemble result.

### 3.4. CDBN Feature Extraction

Convolutional restricted Boltzmann machines (CRBMs) [39,40] are the building block of CDBN and CRBM is an extension of RBM [41] to a convolutional setting. CDBN is an unsupervised hierarchical generative model for feature extraction from unlabeled visual data [42] or acoustic signal [43] using layer-wise greedy bottom-up approach. Inspired by these studies, we construct a five-layer CDBN architecture, as shown in Figure 7. The spectral whitening in frequency domain is performed on fixed sized mel spectrogram before feeding into CRBM and the extracted features from each layer are the input to corresponding higher layer of the network. The filters of various sizes, namely [3 × 7], [3 × 3], [3 × 3], [3 × 3], and [3 × 3] with a fixed pooling size of [2 × 2] are used for the consecutive five layers of CDBN, respectively, where the size is denoted as [frequency bins × frames].

The CDBN is solely trained on cat sound data, and the features from each layer are saved for further classification. The feature map of each layer is divided for FDAP and then the feature vectors are concatenated to feed into classifiers. The CDBN features are divided in frequency range with overlapping. The first three layers is divided into four equal bands, and the fourth and fifth layer features are divided into three and two bands respectively. The concatenation of 50 features from each band of the layers results in 1 × 850-dimensional feature vector. This feature vector is the input to five classifiers, and the prediction results are ensembled with equal priority to get the final predictions by voting. The learned features of CDBN are illustrated in Table 1.

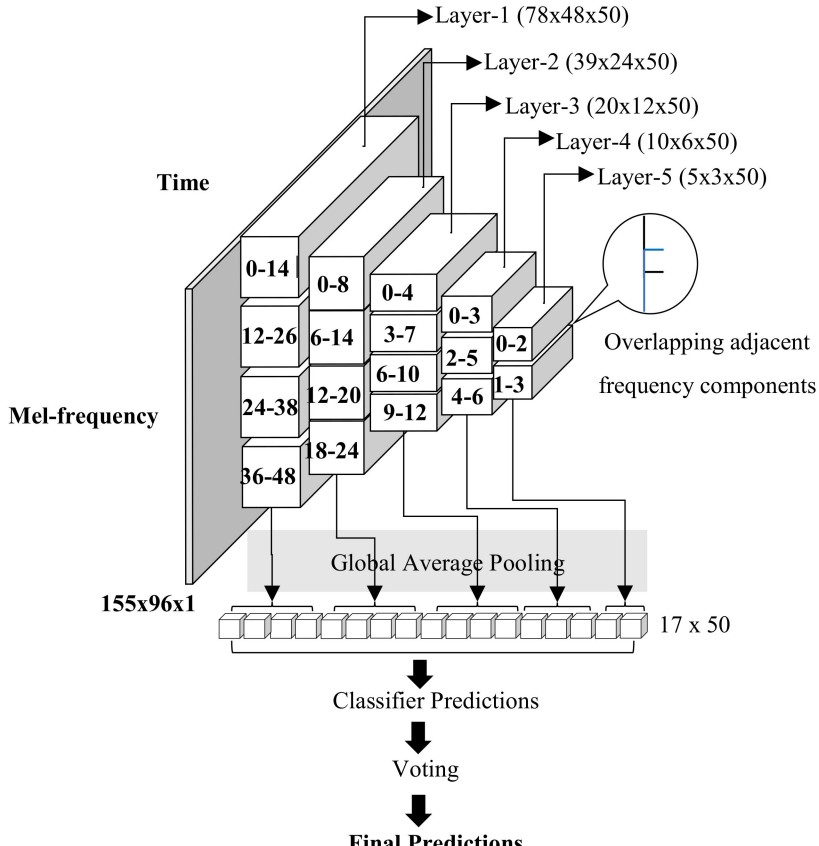

**Figure 7.** An overview of feature extraction from CDBN network and classification. The features from each layer are extracted using FDAP with overlapping shown in each segmented feature block. Each layer parameters are represented as (*time × frequency × channel*). The concatenated feature vector is fed into various classifiers and voting for the predicted probability of the final ensemble result.

**Table 1.** First channel learned feature visualization of proposed CDBN in each layer. The wave form representation of same sample data is in Figure 2.

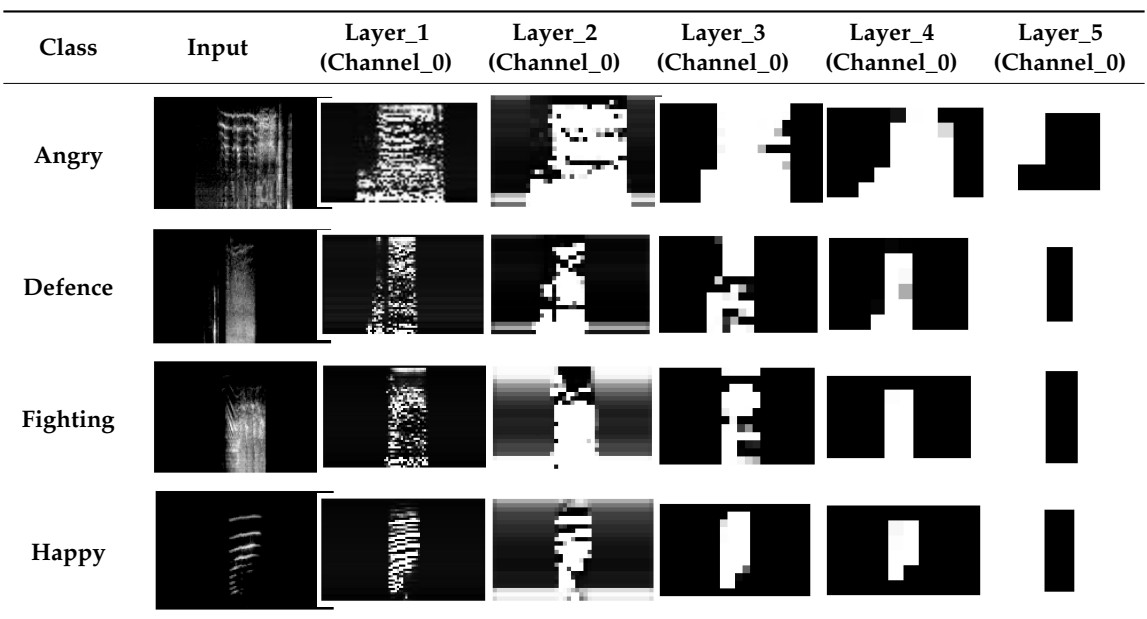

| Class | Input | Layer_1 (Channel_0) | Layer_2 (Channel_0) | Layer_3 (Channel_0) | Layer_4 (Channel_0) | Layer_5 (Channel_0) |
|---|---|---|---|---|---|---|
| Angry | | | | | | |
| Defence | | | | | | |
| Fighting | | | | | | |
| Happy | | | | | | |

**Table 1.** *Cont.*

| Class | Input | Layer_1 (Channel_0) | Layer_2 (Channel_0) | Layer_3 (Channel_0) | Layer_4 (Channel_0) | Layer_5 (Channel_0) |
|---|---|---|---|---|---|---|
| Hunting Mind | | | | | | |
| Mating | | | | | | |
| Mother Call | | | | | | |
| Paining | | | | | | |
| Resting | | | | | | |
| Warning | | | | | | |

### 3.5. Cat Sound Classification

We select five classifiers provided in python sklearn (http://scikit-learn.org) library to classify cat sounds using the learned feature of pre-trained CNN and unsupervised CDBN. These machine learning algorithms are powerful and they can operate well, even with the limited number of classes labeled data. Here, we give a short introduction of classifiers used in this work. *Random forest* (RF) [44] classifier uses majority voting scheme (*Bagging* [45] or *Boosting* [46]) to make the final prediction from a pre-specified number of decision trees. *K-nearest neighbor* (KNN) [47] classifier finds the class to which an unknown object belongs, by using majority voting of *k*-nearest neighbors. Extremely randomized trees (or Extra Tree) [48] classifier tries to find the optimal cut-point on entire given feature randomly. The linear discriminant analysis (LDA) [49] is an extension of Fisher's Discriminant Analysis [50] idea on situation of any number of classes and uses matrix algebra devices to compute it. LDA is a type of Bayesian classifiers that requires an assumption of equal variance-covariance matrices of the classes. Support vector machine [51] (SVM) classifier with a radial basis function (RBF) is used in this research to classify our extracted features non-linearly. The comparative results of these classifiers are mentioned in Section 4.

The ensemble is a well-known machine learning technique that combines the prediction power of multiple classifiers and then classify new data points by taking a (weighted) vote of their predictions, as experimentally proved in [52] for audio classification purpose. A number of recent algorithms have been developed, such as bagging, bucket of models, stacking, and boosting. The ensemble method combines several machine learning techniques into one predictive model in order to decrease variance (bagging), bias (boosting), or improve predictions (stacking) [53]. In this experiment, we select the simplest majority voting with equal priority to ensemble our five-classifiers.

## 4. Results

To evaluate the performance, accuracy, F1-score [54], and area under the receiver operating characteristic curve (ROC-AUC) scores [55] have been employed. The accuracy refers to the percentage of correctly classified unknown data samples, and *F1*-score computes harmonic mean between precision and recall [56]. The ROC-AUC score is measured from each classifier in the receiver operating characteristic (ROC) curve [55]. The ROC curve is used to visualize and analyze the performance of each classifier according to the various decision thresholds associated with it. The Confusion matrix shows the precise performances for the different classes.

In the experiment, we have used 10-fold validation [57] to evaluate the performances according to the diverse configurations of the experimental setting. Then we have compared the performances with respect to the following factors; the amount of data increased by augmentation, the learned features from pre-trained CNN or unsupervised CDBN, conventional GAP or FDAP, the five different machine learning algorithms, and an ensemble of them.

The evaluation results show that the influence of augmentation on the dataset, FDAP method, and majority voting on classifier predictions boost the overall performance and reduce the rate of confusion as we expected. Table 2 illustrates the best performing results of various classifiers using two kinds of learned features with the 3x_Aug dataset.

**Table 2.** Best performances of various classifiers on CNN and CDBN feature extracted from 3x_Aug dataset with FDAP. Bold numbers mark the highest score of the algorithm(s) for learned features from network.

| Classifiers | Accuracy (%) | | F1-Score | | AUC Score | |
|:---:|:---:|:---:|:---:|:---:|:---:|:---:|
| | CNN (±SD) | CDBN (±SD) | CNN | CDBN | CNN | CDBN |
| RF | 84.64 (0.02) | 86.32 (0.03) | 0.85 | 0.86 | 0.988 | 0.988 |
| KNN | 81.60 (0.02) | 80.32 (0.02) | 0.82 | 0.80 | 0.898 | 0.890 |
| Extra Trees | 83.80 (0.03) | 84.29 (0.03) | 0.84 | 0.84 | 0.987 | 0.985 |
| LDA | 78.48 (0.02) | 81.84 (0.03) | 0.78 | 0.82 | 0.975 | 0.979 |
| SVM | 87.43 (0.03) | 90.88 (0.04) | 0.87 | **0.91** | 0.992 | 0.994 |
| Ensemble | **90.80** | **91.13** | **0.91** | **0.91** | **0.994** | **0.995** |

SD—Standard Deviation in each fold of cross validation.

Note that the learned features from the pre-trained CNN with music have produced the comparable results with those from unsupervised CDBN that is solely trained on the cat sound data. The SVM classifier also appears good performance on CDBN features, but the AUC score does not exceed that of the ensemble classifier.

The results of ensemble classifier with or without augmentation, and implementation of GAP or FDAP are presented in Table 3. Note that the performances of classifiers have been consistently improved by increasing the amount of training data with augmentation, regardless of the neural networks that are used for feature extraction. In addition, FDAP provides better performances than conventional GAP in every comparison, which proves that FDAP is more effective for the cat sound classification. The average increase in prediction accuracy on the cat sound dataset using FDAP in CNN is 7.37 and 10.05 in CDBN when 3x_Aug dataset is used for training.

Table 4 shows the confusion matrix of our best performing ensemble classifier using CDBN learned feature with 3x_Aug dataset. We have examined the advantages of using FDAP method over conventional GAP by this confusion matrix. The off-diagonal numbers represent the number of cat sounds in each class that are misclassified by the ensemble classifier. After having analyzed each confusion matrix of the five classifiers and an ensemble classifier, we can reach some conclusions. The Defense ("*Hissing*") and HuntingMind ("*trilling* or *chatting*") sounds are easily distinguishable from other cat sounds at least in our CatSound dataset, so that these classes are relatively less confusing. On the other hand, Happy ("*meow-meow*") and Paining ("*miyoou*"), have some similarity in sound

features so that all classifiers decide these classes more confusing. Likewise, warning ("*growling*") is easily confused with Mating ("*gay-gay-gay*").

**Table 3.** The accuracy, F1-Score and area under ROC curve comparison of CNN and CDBN feature using ensemble classifier on original and augmented datasets. The advantage of FDAP over GAP in each layer feature map of two networks using various datasets are denoted by "GAP" and "FDAP" at the end of corresponding dataset. Bold numbers mark the best-performing algorithm(s) for a dataset using FDAP in a learned feature of the network.

| Cat Sound Dataset | Accuracy (%) | | F1-Score | | AUC Score | |
|---|---|---|---|---|---|---|
| | CNN | CDBN | CNN | CDBN | CNN | CDBN |
| Original_GAP * | 70.71 | 76.09 | 0.690 | 0.760 | 0.958 | 0.963 |
| Original_FDAP # | 79.12 | **82.15** | 0.780 | **0.820** | 0.974 | **0.982** |
| 1x_Aug_GAP | 80.61 | 76.39 | 0.810 | 0.760 | 0.977 | 0.970 |
| 1x_Aug_FDAP | 86.51 | **87.52** | 0.860 | **0.880** | **0.989** | **0.989** |
| 2x_Aug_GAP | 81.89 | 76.60 | 0.820 | 0.760 | 0.982 | 0.969 |
| 2x_Aug_FDAP | **89.31** | 87.96 | **0.890** | 0.880 | **0.993** | 0.991 |
| 3x_Aug_GAP | 83.04 | 79.49 | 0.830 | 0.790 | 0.985 | 0.977 |
| 3x_Aug_FDAP | 90.80 | **91.13** | **0.910** | **0.910** | 0.994 | **0.995** |

\* GAP—Globalaverage pooling; # FDAP—Frequency division average pooling.

**Table 4.** The confusion matrix of the best performing ensemble classifier on 3x_Aug dataset using CDBN features with GAP and FDAP. The first number represents the number of confused cat sounds using GAP and second using FDAP, illustrated as GAP/FDAP format.

| | Angry | Defense | Fighting | Happy | Hunting Mind | Mating | Mother Call | Paining | Resting | Warning |
|---|---|---|---|---|---|---|---|---|---|---|
| Angry | **83 */94 #** | 2/– | 2/– | –/– | –/– | 2/1 | 1/– | 4/1 | –/– | 7/4 |
| Defense | 1/– | **92/97** | –/– | –/– | 4/1 | 1/3 | 1/– | –/– | –/– | 1/– |
| Fighting | 1/1 | 2/– | **84/90** | 1/– | 7/5 | 1/1 | –/2 | 3/1 | –/– | 1/1 |
| Happy | 4/2 | 4/2 | 3/– | **74/90** | 5/1 | 1/– | 1/1 | 1/1 | –/– | –/– |
| HuntingMind | 1/– | 4/– | 4/1 | –/– | **78/96** | 6/1 | –/– | –/– | 2/– | 6/2 |
| Mating | 2/2 | –/– | 2/1 | –/2 | 7/3 | **76/85** | 1/– | 2/2 | 2/– | 8/5 |
| MotherCall | 1/– | –/– | 2/– | 4/2 | 4/2 | 3/1 | **78/94** | 4/– | 5/2 | –/– |
| Paining | 7/5 | 1/– | 3/1 | 8/4 | 1/2 | 1/– | 4/3 | **75/85** | –/– | –/– |
| Resting | –/– | 3/3 | 1/– | –/– | 6/3 | 4/– | 1/– | –/– | **78/92** | 7/1 |
| Warning | 5/2 | 5/2 | 2/– | 1/2 | 4/2 | 5/2 | –/– | 1/1 | 2/1 | **76/89** |

\* Number of cat sounds fall in same or other class using GAP; # Number of cat sounds fall in same or other class using FDAP.

The ROC curves of five classifiers and an ensemble classifier and their corresponding ROC-AUC scores are illustrated in Figure 8. One can observe that the ensemble classifier produces the closest performance to the ideal classifier, even though SVM shows the similar performance.

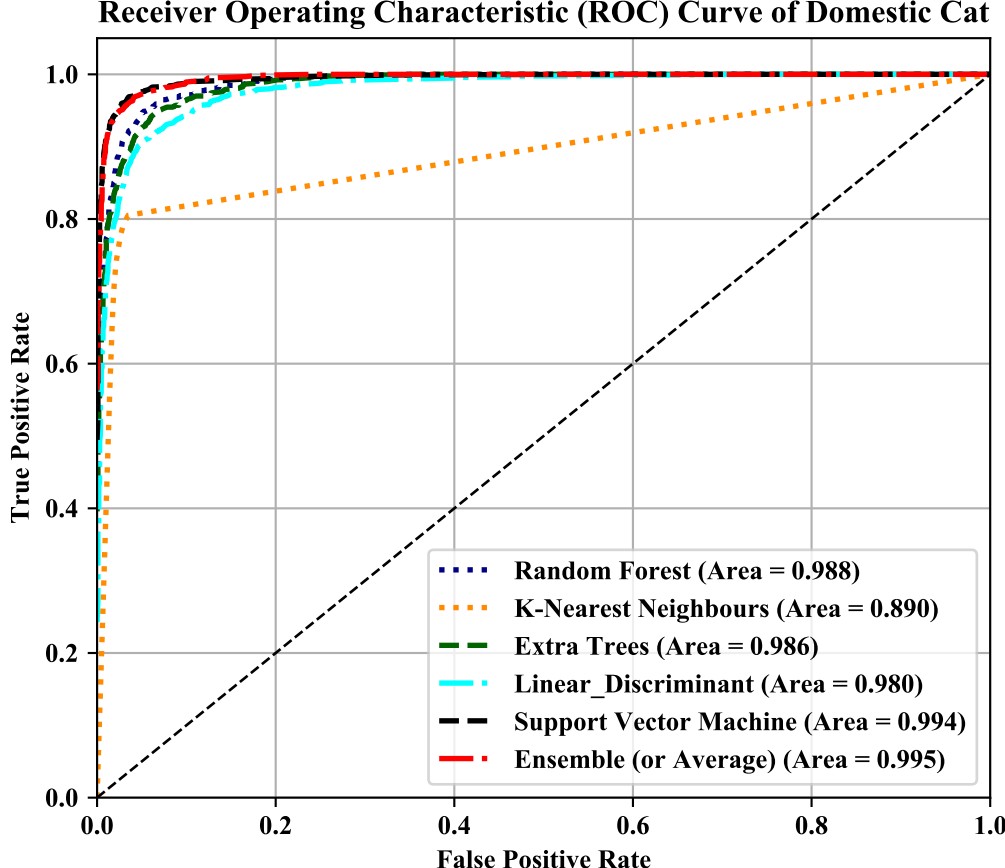

**Figure 8.** ROC curves of the various classifiers with 3x_Aug datasets using CDBN features. Each curve is represented by a unique color and corresponding AUC score is given in parentheses.

## 5. Discussion

The features from the unsupervised CDBN results in better classification accuracy than those from the pre-trained CNN except the case of 2x_Aug_FADP, as shown in Table 2. There might be several reasons. One is that the features from CDBN are solely trained on the cat sounds, while those from CNN are obtained from the pre-trained network with the music data. However, we can conclude that the transfer learning is still valid even though the source and target tasks are slightly different. Another reason is that the FDAP method has been applied to the last three layers of the CNN, while it was used in all the layers of the CDBN. In conclusion, the improvement of the performance in CNN using FDAP is relatively higher than that in CDBN. In future, there is a hope to improve the performance if FDAP is applied to all layers.

The CDBN features may perform better if more training data is available. In the case of classifier comparison, the SVM classifier predicts more accurately than the other classifiers, but it needs relatively more training time. As mentioned earlier, the ensemble classifier achieves the best accuracy in all diverse configurations of experiment setting.

## 6. Conclusions

Domestic cat sound classification is an attempt to achieve better interaction between pet animal and human being. The data-driven approach for sound classification needs large amount of class labeled data. In this work, first we have made a small cat sound dataset with 10 categories and visualized the characteristics of the sound in terms of time and frequency representation. We hope that this dataset could inspire other researchers to study the similar task. We have enhanced the capability of this small dataset by choosing various audio augmentation methods. Also, we modified

the conventional concept of global average pooling (GAP) and used frequency division average pooling (FDAP) for better exploring and the exploitation of learned features of deep neural nets.

In this research, the effect of the data augmentation and FDAP is experimentally studied. Another way to overcome the limited availability of large annotated data for the cat sound classification is the transfer learning. We have found that, even though the source CNN network is trained with music data, it is still beneficial to extract the learned features from small sized cat sound dataset using transfer learning. Unsupervised CDBN architecture is another way to extract the learned feature. The features from the unsupervised CDBN network solely trained on cat sound data performs a little better than those from the CNN trained with music data. Our FDAP method has been applied limitedly only to the last three layers of the CNN, therefore, there is possibility of improvement in performance in future. We have also compared the classification performances of the five different classifiers and an ensemble of them. Finally, we can conclude that the data augmentation, majority voting for ensemble, and FDAP are the ways to boost the classification performance for the small cat sound dataset.

Obviously, our work is also not far from the limitations. The dominant limitation might be lack of expert involvement in labeling the cat sound data, deviation in pre-trained CNN and CDBN, the lack of data, and transfer learning in two different datasets.

In future, there is possibility to improve results if the large labeled dataset is available for end-to-end network training. The various division schemes of feature maps in the layers of a deep neural network could possibly boost the classification performance. In addition, it would be better to take advice from cat sound experts for accurate data labeling. Here, we used the pre-trained CNN with music data, which is somewhat different from cat sound. If there had been any pre-trained network with the animal sound similar to cat sound, the transfer learning could have produced better classification results. The cat sound signal analysis using another type of machine learning algorithm, such as extreme learning machine (ELM) or multi-kernel ELM (MKELM) [58], are possible techniques to make more comparative study in future.

**Author Contributions:** Conceptualization, Y.R.P. and J.L.; Data curation, Y.R.P.; Methodology, Y.R.P. and D.K.; Project administration, Y.R.P.; Resources, J.L.; Writing—original draft, Y.R.P.; Writing—review & editing, J.L.

**Funding:** The National Research Foundation of Korea (NRF) support us under Basic Science Research Program (NRF-2015R1D1A1A01058062). The Institute of Energy Technology Evaluation and Planning (KETEP) support under Energy Efficiency and Resources Core Technology Program (No. 20172510102150). The Chonbuk National university research fund also support this study.

**Acknowledgments:** The research leading to these results, authors would like to thank Korean Ministry of Education and Ministry of Trade, Industry and Energy for their funding.

**Conflicts of Interest:** The authors declare no conflict of interest. The funding sponsors had no role in the design of the study; in the collection, analyses, or interpretation of data; in the writing of the manuscript, and in the decision to publish the results.

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
