# Peer review of "Domestic Cat Sound Classification Using Learned Features from Deep Neural Nets"

_applsci, doi:10.3390/app8101949_

Round 1

Reviewer 1 Report

It is an interesting paper on classification of cat sounds. However the following major issues should be addressed:

1. There is a big overlap (in both text and figures) with another paper of almost the same authors published in International Journal of Fuzzy Logic and Intelligent Systems

Vol. 18, No. 2, June 2018, pp. 154-160

The authors should explain the differences, update the text, clarify if they can use the same figures, etc.

2. It is not clear how the authors annotated the dataset. If i understood correctly, it was gathered from various  internet sources without any expert assistance. How do we know that the dataset is really representative of these classes? Also, is the dataset publicly available so we can examine it?

3. One of the classifiers typically used in sound recognition, i.e. HMM is not used. Such a comparison is mandatory.

The authors should see and cite such papers, for example class-specific and UBM HMM in “A Novel Holistic Modeling Approach for Generalized Sound Recognition”.

4. A minor issue is that the text in general needs polishing, since now it is naive in many points.

5. Why was the specific data augmentation method selected?

6. It is not clear why the authors use a CNN pre-trained on the Million song dataset for classifying cat sounds. Please give a clear motivation.

Author Response

Dear Reviewer, 

We are very thankful to your great comments in our paper “Domestic cat sound classification using learned features from deep neural nets”. We are trying to answer in your valuable comments but due to time limitation, some tasks are postponed for future work. Our response to your great comments is included with this report. 

All the modification as your valuable comments will reflected in modified copy of our manuscript. 

We are always waiting for your valuable comments.

Reviewer 2 Report

See attached file.

Author Response

Dear Reviewer, 

We are very thankful to your great comments in our paper “Domestic cat sound classification using learned features from deep neural nets”. We are trying to answer in your valuable comments but due to time limitation, some tasks are postponed for future work. We attach the response file with this mail.

 All the modification as your valuable comments will reflected in modified copy of our manuscript. 

 We are always waiting for your valuable comments if any.

Reviewer 3 Report

The authors proposed a method using two types of features learned from pre-trained CNN on music data by transfer learning and from CDBN for domestic cat sound classification. Experimental study was performed to validate this method with various classifiers. The addressed issue is interesting and the method is useful. The following revisions are needed before accepted for publication:

1. It would be interesting if the authors can visualize some features learned from the deep neural networks.

2. How about combining these two types of features? Please give a brief discussion on this point.

3. Some related methods especially about computational algorithms should be included in the references, for example: An adaptive neural network approach for operator functional state prediction using psychophysiological data; Multi-kernel extreme learning machine for EEG classification in brain-computer interface; A novel multilayer correlation maximization model for improving CCA-based frequency recognition in SSVEP brain-computer interface; Sparse Bayesian classification of EEG for brain-computer interface; Discriminative feature extraction via multivariate linear regression for SSVEP-based BCI.

4. What are the potential limitations of the method? Please give a discussion on this and also provide several future study directions for addressing these limitations.

Author Response

(The authors gave the same response as above.)

Round 2

Reviewer 1 Report

Most of my comments were addressed in the response of the authors but it is not clear where exactly in the updated manuscript. It would be nice if the authors could clarify explicitly the changes they made.

But most importantly the comparison with HMMs (i.e. my previous comment "Point 3: One of the classifiers typically used in sound recognition, i.e. HMM is not used. Such a comparison is mandatory. The authors should see and cite such papers, for example class-specific and UBM HMM in “A Novel Holistic Modeling Approach for Generalized Sound Recognition”.") is not addressed at all.

Author Response

Dear Reviewer, 

We are very thankful to your great comments in our paper “Domestic cat sound classification using learned features from deep neural nets”. We attach the response file with this mail.

All the modification as your valuable comments are included in modified copy of our manuscript. 

We are always waiting for your valuable comments if any.

Round 3

Reviewer 1 Report

I would like to thank the authors for implementing my comments. I think that the article can be accepted after a thorough proofreading.

Author Response

Dear Reviewer, 

We are very thankful to your great comments in our paper “Domestic cat sound classification using learned features from deep neural nets”. We make English correction in our manuscript from English correction center. All the modification according to your valuable comments are included in  modified copy of our manuscript. 

We are always waiting for your valuable comments if any.